# Automatic time in bed detection from hip-worn accelerometers for large epidemiological studies: The Tromsø Study

**Marc Weitz**[1]*, **Shaheen Syed**[1,2], **Laila A. Hopstock**[3], **Bente Morseth**[4], **André Henriksen**[1], **Alexander Horsch**[1]

**1** Department of Computer Science, UiT The Arctic University of Norway, Tromsø, Norway, **2** Department of Seafood Industry, Nofima AS, Tromsø, Norway, **3** Department of Health and Care Sciences, UiT The Arctic University of Norway, Tromsø, Norway, **4** School of Sport Sciences, UiT The Arctic University of Norway, Tromsø, Norway

* marc.weitz@uit.no

**Data availability statement:** The data of the Tromsø Study cannot be shared publicly to

## Abstract

Accelerometers are frequently used to assess physical activity in large epidemiological studies. They can monitor movement patterns and cycles over several days under free-living conditions and are usually either worn on the wrist or the hip. While wrist-worn accelerometers have been frequently used to additionally assess sleep and time in bed behavior, hip-worn accelerometers have been widely neglected for this task due to their primary focus on physical activity. Here, we present a new method with the objective to identify the time in bed to enable further analysis options for large-scale studies using hip-placement like time in bed or sedentary time analyses. We introduced new and accelerometer-specific data augmentation methods, such as mimicking a wrongly worn accelerometer, additional noise, and random croping, to improve training and generalization performance. Subsequently, we trained a neural network model on a sample from the population-based Tromsø Study and evaluated it on two additional datasets. Our algorithm achieved an accuracy of 94% on the training data, 92% on unseen data from the same population and comparable results to consumer-wearable data obtained from a demographically different population. Generalization performance was overall good, however, we found that on a few particular days or participants, the trained model fundamentally over- or underestimated time in bed (e.g., predicted all or nothing as time in bed). Despite these limitations, we anticipate our approach to be a starting point for more sophisticated methods to identify time in bed or at some point even sleep from hip-worn acceleration signals. This can enable the re-use of already collected data, for example, for longitudinal analyses where sleep-related research questions only recently got into focus or sedentary time needs to be estimated in 24 h wear protocols.

## Introduction

Accelerometers have become a common assessment method to monitor human physical activity. Accelerometers are small body-worn sensors that can monitor subjects for a week or

control for data sharing, including publication of datasets with the potential of reverse identification of de-identified sensitive participant information.Data are available from the Data and Publication Committee for the Tromsø Study (Contact information: The Tromsø Study, Department of Community Medicine, Faculty of Health Sciences, UiT The Arctic University of Norway; e-mail: tromsous@uit.no) for researchers who meet the criteria for access of confidential data. An anonymized copy of the ActiGraph/Oura/Polar validation dataset has been submitted to and will be available from Dataverse (https://doi.org/10.18710/VP5DMU).

**Funding:** The publication charges for this article have been funded by a grant from the publication fund of UiT The Arctic University of Norway.

**Competing interests:** The authors have declared that no competing interests exist.

more without intervention. Due to its non-invasive and unobtrusive application, accelerometers are used in many population [1–3] and clinical studies [4,5]. Besides measuring physical activity, accelerometers worn on the wrist are also frequently used to measure sleep under free-living conditions [6,7], often referred to as actigraphy [8]. Compared to the gold-standard method of sleep assessment, polysomnography (PSG), actigraphy is considerably cheaper, does not require the participant to stay the night in a sleep laboratory and can easily be applied over several days. Depending on the specific study design, many large-scale epidemiological studies using accelerometers can, by that, obtain sleep estimates even without additional costs, as accelerometer surveillance is already part of the protocol.

In fact, accelerometers have been utilized to detect and measure sleep or time in bed (TiB) periods over the last decades. Early algorithms, e.g. proposed by Cole et al. [9] and Sadeh et al. [10], used heuristic algorithms to discriminate sleep epochs from sedentary behavior based on the number and sequence of counts per minute measured on the participants' dominant and non-dominant wrist. Counts are a proprietary unit developed to summarize the actual raw acceleration signal measured in gravitational units ($1\,g = 9.806{,}65\,\mathrm{m\,s^{-2}}$) on a given amount of time, called epoch. For a comprehensive description on how counts are calculated, see Neishabouri et al. [11].

In the recent years, the research focus has shifted to machine learning (ML) based stochastic algorithms. Palotti et al. [12] conducted a large benchmark study in which they compared the performance of different ML methods on the count data of the Multi-Ethnic Study of Atherosclerosis (MESA) sleep dataset [13,14]. Among all tested models, Long-Short Term Memory networks (LSTMs) were found to commonly performing best. LSTMs are a type of neural network models that have been proposed and validated to perform well on sequential data [15]. While a few openly available count datasets, like the MESA study, have been published, available raw data is scarce.

Algorithms using raw data have become popular in the last decade, due to the higher information content of the signal and a better inter-sensor comparability [16]. Van Hees et al. [17, 18], for instance, derived manually engineered features, such as the arm angle, from the raw signal and used them to detect sleep with and without an additional sleep diary from wrist-worn accelerometers. The dataset employed in these studies is one of the few openly available datasets containing raw acceleration and PSG data [19]. Later approaches also examined the potential of ML to detect sleep periods on this dataset by using Random Forests on similar, manually engineered features [20].

However, while many good algorithms have been proposed and validated for wrist-assessment over the last two decades, none has been proposed for hip-placement on an adult population [21]. Simultaneously, accelerometer data analysis is sensitive to the wear location, and methods developed for hip placement are usually not applicable to data collected from wrist placement, and vice versa, as the acceleration profiles vary significantly [22]. The reasons why no algorithm has been proposed yet for hip-placement are manifold. On one hand, actigraphy has since its establishment had a strong focus on the wrist-placement, as wrist-placement has been considered less obtrusive and thus leads to fewer non-wear days [23]. Concurrently, it was also argued that the use of wrist-specific features yields better discriminability on data collected from the wrist than from the hip [17,18,20]. Hip-placement, on the other hand, has its origins in the measurement of physical activity, as hip-placement has been shown to provide superior estimates of physical activity when compared to wrist-placement [24]. To decrease study participant burden, many early study protocols comprised the removal of the device during sleep. However, studies indicate that this leads to a lower

overall study protocol adherence [25], and a considerable number of studies using hip placement are now following 24 h wear protocols to circumvent this problem. It is also worth noting that sleep or TiB usually have not been the primary variables of interest in these studies, but can nevertheless provide meaningful additional information, or can be used to, for example, differentiate sedentary time.

For that reason, this study aims to provide a new algorithm to discriminate between TiB and time out of bed in data from hip-worn accelerometers. Our goal with this is to provide a tool for studies that use hip-based accelerometry, e.g., because their primary focus is on physical activity. Additionally, our algorithm can be used for long-running studies that have data from previous data acquisition rounds that were collected on the hip and who want to get additional insights.

## Materials and methods

### Datasets

In this study, we only use secondary data obtained from the respective data owners. All data collection had been approved by the Regional Committee for Medical Research Ethics (REC North ref. 2014/940 and ref. 2019/557) and the Norwegian Data Protection Authority, and all participants gave written informed consent. Details of study designs and data collection is found elsewhere [3,26]. Data collection was carried out in accordance the Declaration of Helsinki. The usage of the Tromsø Study data has been approved by the Data and Publication Committee of the Tromsø Study (DPU 50/21) on 05.01.2021. The ActiGraph/Oura/Polar dataset has been retrieved on 14.06.2023. The authors did not have access to information that could identify individual participants. A brief overview over the datasets is also provided in Fig 1.

**Tromsø Study TiB dataset** The Tromsø Study TiB dataset consists of recordings of 579 participants from the seventh survey of the Tromsø Study (Tromsø7) collected 2015-2016 [3]. The participants wore an Actiwave Cardio (CamNtech Ltd, Cambridge, UK) glued to the participants chest, and an ActiGraph wGT3X-BT (ActiGraph, Pensacola, FL) worn on an elastic band on the right hip simultaneously for approximately 24 h. While the ActiWave Cardio was removed after this time, due to its limited battery capacity, the ActiGraph remained in place

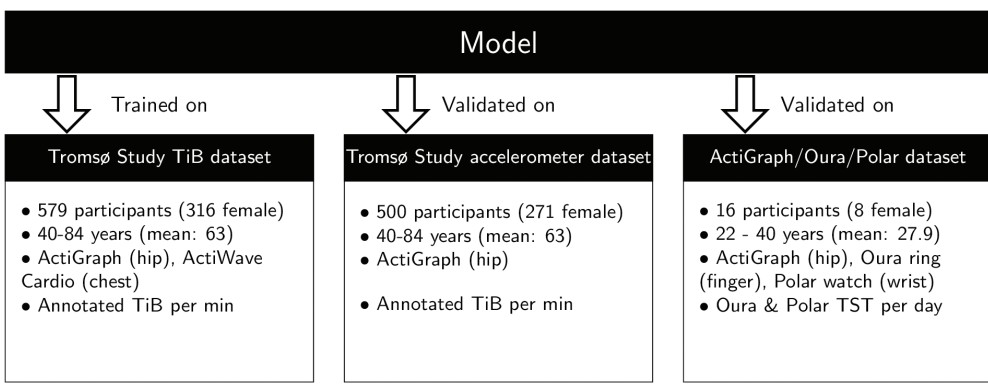

TiB: Time in Bed; TST: Total Sleep Time

**Fig 1. Overview over the different datasets and how they were used in this study.** For each dataset, the key demographic features, which sensors were used and the target variable on which the model was evaluated are shown.

and was worn for seven consecutive days in total. The TiB labels were derived from the Acti-Wave Cardio by manual labeling according to the manufacturer's manual by the authors M.W. and S.S., independently. Both annotators were instructed to identify the time the participant went to bed and the time the participant got out of bed again. The time the participant went to bed was defined by a drastic change in the activity pattern from activity to barely no activity and a position change from an active to a lying position. The time the participant got out of bed again was defined by a longer preceding episode of low activity followed by a higher amount of activity and a change of position from a lying to an active position. To ensure data quality, only participants with an intersection over the union (IoU) score of over 0.9 between the two annotators' ratings were included in this study. The IoU score was calculated by dividing the total time that both annotators labeled as TiB (the intersection of both ratings) by the total time that at least one annotator labeled as TiB (the union). For all participants meeting this requirement ($n = 444$), the intersection of the two annotators' rating was labeled as TiB and aligned with the corresponding data of the ActiGraph sensor. Examples of this procedure can be found in the supplementary material S1 Fig.

**Tromsø Study accelerometer dataset** The Tromsø Study accelerometer dataset was sampled from all participants wearing an ActiGraph in Tromsø7. The motivation for this dataset was to provide additional validation for the model trained on the TiB dataset. However, compared to the TiB dataset the labels of this dataset rely only on the ActiGraph monitor as all data for which ActiWave Cardio data was available was already used to construct the TiB dataset [3,27]. Besides that, the same labeling procedure as for the TiB dataset was applied.

From the around 6000 participants who wore the accelerometer for seven consecutive days (around 42.000 days in total), we randomly sampled 500 24 h-recordings (noon to noon). This sample size was chosen to approximately match the size of the TiB dataset and as 500 seemed to be a feasible number of days to annotate manually. Each participant was maximally included once in this subset. These 500 recordings were again labeled by M.W. and S.S. independently with the same IoU criteria of over 0.9 applied as in the previous dataset. For all participants meeting this requirement ($n = 465$), the intersection of the two annotators' rating was labeled as TiB and aligned with the corresponding data of the ActiGraph sensor.

**ActiGraph/Oura/Polar validation dataset** The ActiGraph/Oura/Polar dataset [28] was originally collected as part of a device comparison study [26]. In the original study, 21 participants wore an Actigraph on their right hip together with two consumer wearables (a Polar Vantage [Polar Electro Oy, Kempele, Finland] watch and an Oura ring [Oura Health Oy, Oulo, Finland]). Five participants had to be excluded, because they removed the ActiGraph device during sleep. For the remaining 16 participants we obtained total sleep duration estimates from the Polar Vantage and the Oura ring. The motivation of using this dataset in our study was to validate the model also on demographically different data.

## Procedure

**Data preparation** For all datasets, the ActiGraph raw data was downloaded as GT3X files using ActiLife and subsequently loaded using the Python package PAAT [29]. Accelerometer calibration and correction coefficients were estimated using GGIR [30] and the acceleration data calibrated accordingly [31,32]. Subsequently, we normalized the data with the mean and standard deviation of the acceleration per minute for all three axes across all participants of the respective dataset.

**Model training** On the TiB dataset, we trained uni- and bidirectional LSTMs and evaluated their within distribution generalization performance by 10-fold cross-validation. Within each fold, we augmented the training data by randomly cropping out 100 sequences from the

original sequence of random length between 5 min and the entire time series. With a 50% likelihood, the sequence got flipped in order to simulate the sensor being worn upside down. Also with 50% likelihood, the sequence got reversed to simulate inverse turning behavior. Finally, in 10% of the cases a slight noise of a Gaussian distribution with a standard deviation of 0.1 was applied to increase robustness of the trained models. From the trained model we selected the best model based on performance (accuracy) and sparsity for external validation and the application to free-living data.

**Model evaluation** To test the out-of-distribution performance of the model, we applied the model to two additional datasets: Another subset of the Tromsø Study accelerometer dataset which had not been used for training and the ActiGraph/Oura/Polar validation dataset. For both datasets, we applied the same preprocessing steps as mentioned earlier. We used the selected model to predict TiB and wake periods on both datasets. On the Tromsø Study accelerometer dataset, we compared the model's predictions with the respective labels. Performance metrics were calculated using the standard functions from the Scikit-Learn Python package. As for the ActiGraph/Oura/Polar validation dataset only total sleep duration data was available, we calculated the total TiB from the predictions of our model on all fully recorded days. This usually excluded the first and last recording day, when the participants received or returned the ActiGraph, as comparing TiB estimates for these days is difficult. The daily estimates were averaged per participant and compared with the estimates from the Polar Vantage and the Oura ring, respectively.

**Non-wear time detection** Additionally, we calculated non-wear time (NWT) with the method of Van Hees et al. [33,34] to analyze potential influence of NWT on the TiB estimates. This method calculates NWT based on the standard deviation of the raw data. An interval is considered NWT if the standard deviation is less than 3 mg, or if the value range is less than 50 mg for at least 2 of the 3 axes, respectively. Initially, the interval length was set to 30 min. However, we followed later recommendations to use a 60 min interval with a 15 min sliding window which has been shown to decrease incorrectly classifying sedentary episodes as NWT [34].

## Results

### Tromsø Study TiB dataset

A hyperparameter search over number of layers (1, 2, and 4) and number of LSTM cells per layer (1, 2, 4, 8, 16, 32, 64, and 128) revealed accuracies up to 94% for unidirectional and up to 95% for bidirectional LSTMs, with generally low standard errors across the 10 iterations of cross-validation. A detailed overview over the hyperparameter search results can be found in the supplementary material S2 Table. Based on the obtained results, we selected a bidirectional LSTM with 4 layers and 32 cells per layer for the further evaluation.

### Tromsø Study accelerometer dataset

As shown in Fig 2A, the model achieved an accuracy of 92.0% on all 465 days, with a baseline accuracy (achieved by always responding "out of bed") at 65.7% (dotted line), a sensitivity (true positive rate) of 96.6% and a specificity (true negative rate) of 89.4%. In total, 12 days had a predicted TiB of less than one or more than 23 h and were therefore considered as outliers (Green triangles in Fig 2B). These 12 days, of which 11 days were either fully labeled as TiB (9 days) or the model predicted no TiB at all (2 days), accounted for a loss of ~1% overall accuracy. Upon exclusion, the model reached an accuracy of 93.3%, a sensitivity of 96.9% and specificity of 91.4%. Overall accuracy did not increase further when applied only on days

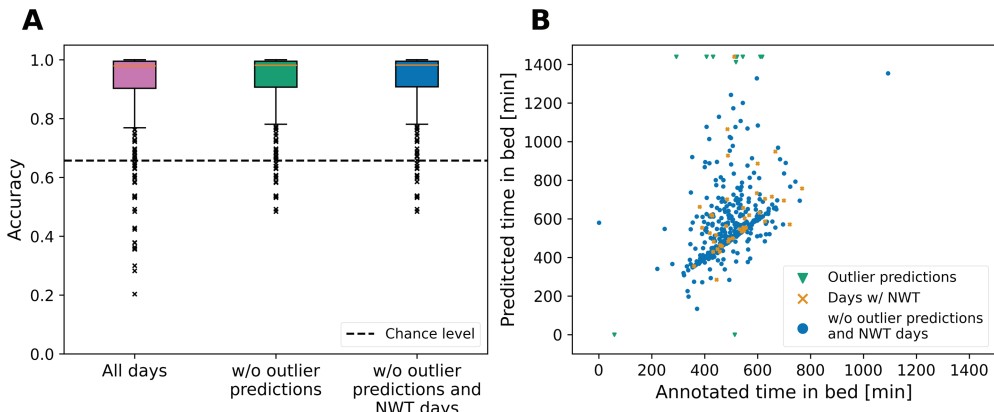

**Fig 2. Validation results on the Tromsø Study accelerometer data subsample.** (A) Accuracies for the same criteria as for the ActiGraph/Oura/Polar dataset. (B) Predicted times in bed plotted against the annotated TiB. The selected model reached an accuracy of over 90% on the additional, unseen data from the Tromsø Study and up to 95.6% when removing unreasonably long predictions (outliers of less than one or more than 23 h of predicted time in bed).

without NWT. Compared to all days, the accuracy increased by 0.12% to 92.12% by excluding 40 days containing NWT. When the outlier criterion is applied, accuracy increased by 0.1% to 93.42% while excluding 39 days more. A more comprehensive overview over various metrics is provided in S5 Table.

## ActiGraph/Oura/Polar validation dataset

On average, our algorithm estimated a total TiB of 633 min ($SD$ = 238 min) per participant and day. The Polar Vantage watch measured a similar total sleep duration of 639 min ($SD$ = 105 min) while the Oura ring measured a shorter sleep duration of 415 min ($SD$ = 56 min). For one of the 16 participants of this dataset, our model predicted almost all time as TiB. When excluding the data of this participant, the estimated total TiB per participant and day of our model was 584 min ($SD$ = 134 min).

## Discussion

Our study showed that TiB can be inferred reasonably well from hip-worn accelerometer data using the proposed method across demographically different datasets. This can enable further analyses of existing and prospective data collected from the hip like from the Tromsø Study [3,27,35] or the German National Cohort Study [36]. However, our method only utilized manually labeled data instead of more objective measures such as cameras or bed sensors. It should, thus, rather be seen as a replacement of manual labeling unless additional more objective validation has been obtained. Nevertheless, it is worth to note that manual labeling has been shown to be a viable option to annotate TiB [37]. It remains also an open question to which extent our results generalize to the detection of sleep, which we could not test within the scope of this study as no appropriate data from hip-worn accelerometers together with annotated sleep times is currently openly available.

We included several steps in this study to ensure generalization as much as possible. However, the lack of openly available datasets also prevented us from additional objective bench marking. We believe such a dataset would be highly appreciated as some recent methods have been shown to largely overfit and thus not being able to keep up with their claims [38].

It is worth noting that the situation for wrist-worn accelerometers is only slightly better, but at least one open dataset [19] exists which has been commonly utilized in similar studies [17,18, 20].

An important question in accelerometer data analysis is the temporal resolution on which the data is analyzed. Modern accelerometers often collect raw data at a sampling frequency of 100 Hz or higher. However, even though a common mantra in machine learning is "the more data, the better", we found a resolution of one data point per minute to be satisfactory for this task. One reason for this was that LSTMs model time-dependencies on the level of data points and not on absolute time. As movement under sleep is rather static, we decided for a lower resolution to allow the model to capture the long term dependencies. It is worth to note that, in fact, other researchers successfully employed high-resolution acceleration signals in a spectral analysis approach to discriminate sleep from being awake [39]. However, employing a higher resolution in a machine learning approach costs significantly more computation time to train than in a spectral analysis. Other researchers have been using features derived from the high-resolution raw data to tackle this problem [20], but no definite answer which features or resolution is ideal has been given.

Another important issue is the comparability between different measurement methods. A method that has become increasingly popular over the last years are consumer wearables such as smart watches [40,41]. Consumer wearables offer a cheap solution for continuous recording, are easy to apply and already many people own already such a device by themselves. Our results indicate that our algorithm produces similar daily estimates like common smart watches like the Polar Vantage. However, the total sleep time (TST) estimates of the Polar Vantage of 639 min (10.65 h) were lying far above the sleep duration one would expect for this age group. With 415 min (6.92 h), the Oura ring had even a little bit less estimated TST than expected, but provided more reasonable results. One reason for this might be the small sample size of the used dataset. Nevertheless, given these limitations, our results seem comparable to the once obtained from common consumer variables. Further, we could not find any systematic distortions for the Polar Vantage, but a trend to higher over-estimation of our model compared to the Oura Ring (see S3 Fig. in the supplementary material). This was partially surprising as our model aims to measure TiB and, thus, should provide longer duration than TST.

A crucial strength of this paper is to provide an opportunity to estimate TiB from already collected data. With this, we anticipate the chance for longitudinal analyses of sleep and TiB-related research question in settings where these questions only recently got into focus, but prior data exists. One example for such a setting might be already long-running cohort studies (such as the Tromsø Study) for which ActiGraph data exists from previous iterations, but which might be extended for further research questions and areas in the future. It is important to note that our research does not aim to act as a primary tool of sleep or TiB assessment. In that case, the reader might be referred to the intensive body of scientific literature that has investigated assessing sleep from wrist-worn accelerometers. However, retrospective analyses of sedentary time in 24 h wear protocols can be a promising use case for this method.

Another important contribution of this study is the introduction of accelerometer-specific data augmentation as a tool to improve model generalization. Data augmentation has become an important step across all major machine learning application domains such as computer vision or natural language processing. The use of machine learning in accelerometry research is increasing, so improving our understanding on how we should augment the data in this process is a necessity. In this study, we proposed a heuristically motivated set of augmentation strategies that are intended to model certain aspects like upside down placement of the accelerometer, reversed movements and additional noise. While this study could not provide

quantitative estimates of the effect sizes of the augmentation methods, we believe that we were able to provide a starting point for more sophisticated and empirically tested augmentation methods in the future.

Despite the promising results, it is still important to highlight that the validation of our model is limited. We tried to validate its generalization performance by all available means. However, as appropriate data in this field, like the data we used in this study, is only rarely openly available, the options to validate our model were limited as well. To overcome this problem, the used model is included in the Python package PAAT [29] to ease validation by external researchers. Note, that the model, which is trained with Tensorflow, can easily be adapted and used also with R. While training the model took considerable time, applying the model to a seven day recording takes only approximately 1.2 s. An overview over how long the different processing steps take, is provided in the supplementary material S4 Fig.

In conclusion, we provide an algorithm to identify TiB episodes from hip-worn accelerometers. The purpose of this algorithm is mainly to support the identification of sleep especially in retrospective analyses of already collected data in epidemiological studies. The limited available data for this study suggests that the algorithm generalizes well and produces comparable results to consumer wearables, a tool increasingly used in free-living studies. However, validation is still limited and needs further research in the future.

## Supporting information

**S1 Fig. Examples of the TiB annotation procedure and results.** Green and red dotted lines represent the annotations of the two annotators. All other information were visible to the annotators during annotation. (A) Both annotators labeled bed and rise time very similarly (high IoU score) and jointly overruled the suggested rise time (dotted line). (B) Both annotators labeled rise time very similar, but had some disagreement on the bed time (lower IoU score than (A), but still high). While annotator 1 agreed with the suggested bed time, annotator 2 labeled a later onset. (C) Both annotators agreed on the suggested rise time, but labeled the bed time very differently (low IoU score). IoU scores were calculated for all annotations by dividing the intersection (time labeled as in bed by at least one of the annotators) by the union (time labeled as in bed by both annotators).
(PDF)

**S2 Table. Within-distribution training results obtained by 10-fold cross-validation for uni- and bidirectional LSTMs on the Tromsø Study sleep dataset.** The models reached accuracies up to 94% for unidirectional and up to 95% for bidirectional LSTMs with generally low standard errors across the 10 iterations of cross-validation. Based on the obtained results, we selected a bidirectional LSTM with 4 layers and 32 cells per layer for the further evaluation.
(PDF)

**S3 Fig. Bland-Altman plots for the predictions of our model compared to the total sleep duration measured by the Polar Vantage and Oura ring.** (A) No systematic distortion is visible for the Polar Vantage. (B) The estimates of our model seem to overestimate the Oura ring's total sleep duration the longer the sleep/TiB is.
(PNG)

**S4 Fig. Processing time for the different pipeline steps.** Application of the TiB algorithm is very quick even if the resampling of the acceleration data is taken into account. Reading the GT3X file is usually the slowest step. Processing times were measured on a desktop PC (Intel Core i9-10850K processor with a 64-bit instruction set, 128GB RAM and an Asus GeForce

RTX3090 GPU) and a laptop (Intel Core i7-1165G7 processor with 32GB RAM and a 64-bit instruction set).
(PNG)

**S5 Table. Overview over various classification metrics for the Tromsø Study accelerometer dataset.** All metrics were calculated on subject-level using the respective functions of the Scikit-Learn Python package. The table shows between-subject averages and standard deviations (in brackets).
(PDF)

**S6 Table. Target and outcome distributions for the Tromsø Study accelerometer dataset.** 1st, 10th, 25th, 50th, 75th, 90th and 99th quantiles are shown for the three criteria shown in Fig 2, respectively.
(PDF)

## Acknowledgments

This work was supported by the High North Population Studies at UiT The Arctic University of Norway. The authors would like to thank Dilip K. Prasad for his feedback on earlier versions of this work.

## Author contributions

**Conceptualization:** Marc Weitz.

**Data curation:** Shaheen Syed, Laila A. Hopstock, Bente Morseth, André Henriksen, Alexander Horsch.

**Methodology:** Marc Weitz.

**Project administration:** Marc Weitz.

**Software:** Marc Weitz.

**Supervision:** Alexander Horsch.

**Writing – original draft:** Marc Weitz.

**Writing – review & editing:** Marc Weitz, Shaheen Syed, Laila A. Hopstock, Bente Morseth, André Henriksen, Alexander Horsch.

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
