## [Decision Letter · Decision Letter 0]

26 Jun 2024

PONE-D-24-15603Automatic time in bed detection from hip-worn accelerometers for large epidemiological studies: The Tromsø StudyPLOS ONE

Dear Dr. Weitz,

Thank you for submitting your manuscript to PLOS ONE. After careful consideration, we feel that it has merit but does not fully meet PLOS ONE’s publication criteria as it currently stands. Therefore, we invite you to submit a revised version of the manuscript that addresses the points raised during the review process.

We look forward to receiving your revised manuscript.

Kind regards,

Vitor Barreto Paravidino

Academic Editor

PLOS ONE

3. In the online submission form, you indicated that [The data of the Tromsø Study cannot be shared publicly because of the risk of potential reidentification, but can be obtained upon application to the Data and Publication Committee for the Tromsø Study: https://uit.no/research/tromsostudy. The ActiGraph/Oura/Polar validation dataset can be obtained from A.He. (andre.henriksen@uit.no) upon reasonable request.]. 

Reviewers' comments:

Reviewer's Responses to Questions

**Comments to the Author**

1. Is the manuscript technically sound, and do the data support the conclusions?

Reviewer #1: Partly

Reviewer #2: Partly

2. Has the statistical analysis been performed appropriately and rigorously? 

Reviewer #1: No

Reviewer #2: No

3. Have the authors made all data underlying the findings in their manuscript fully available?

Reviewer #1: No

Reviewer #2: No

4. Is the manuscript presented in an intelligible fashion and written in standard English?

Reviewer #1: Yes

Reviewer #2: No

5. Review Comments to the Author

Reviewer #1: This paper presents an algorithm aiming at discriminating time in bed and time out of bed based on hip-worn accelerometry data. While hip-worn accelerometry data focused on physical activity, such an algorithm would enable to reuse this data to get other outputs related to sleep. This algorithm is based on an augmentation data process and machine learning models of type Long-Short Term Memory networks (LSTM). This study has potential because the procedure used well-known algorithms applied in similar contexts and some techniques to add noise or take into account potential misuse of devices (flipped sequence, Gaussian noise), but there are weaknesses that need to be revised before publication.

Plos one questions

1. Is the manuscript technically sound, and do the data support the conclusions?

Partly. Please consider for instance my comments / questions regarding data augmentation, the different methods you used to label the data.

2. Has the statistical analysis been performed appropriately and rigorously?

No. I wanted to answer “partly” but I cannot… Please consider my comments / questions regarding the use of additional indicators for the results section. In my point of view, providing accuracy is good but not enough.

3. Have the authors made all data underlying the findings in their manuscript fully available?

No. Data is not available but could be accessed on “reasonable request”.

4. Is the manuscript presented in an intelligible fashion and written in standard English?

Yes. I am not an English speaker but I think it is clear. Some typos are highlighted at the end of my comments.

General comments

As you mentioned in the paper, wearing hip-worn accelerometers during 24h can be a burden for participants, especially during sleep hours (“wrist-placement has been considered less obtrusive and thus leads to fewer non-wear days”; “To decrease study participant burden, many early study protocols comprised the removal of the device during sleep”). This fact might explain the lack of data appropriate for this kind of study. I think you should discuss the potential of using hip-worn accelerometers in behavior monitoring, maybe in future studies. In addition, it is probably a limit to the study, not so much a scientific limit as a functional one. It should be mentioned.

In your process, you use many types of data but it is unclear from the description how this data is used in the procedure. For instance in the procedure, it is not clear which dataset was used to train the models. Is it the “Tromsø Study TiB dataset”, “the Tromsø Study accelerometer dataset” or a mix of both? Please clarify the function of each dataset you used.

I am a bit skeptical about the use of ActiGraph/Oura/Polar dataset. Only 15 (16 – 1) participants provide “relevant information”. Moreover, it seems there is a confusion between sleep duration and time in bed (please consider my comments / questions related to the procedure section).

You mentioned that “data augmentation” improved the model generalization. How this technique improved the model. There is no comparisons with and without data augmentation. I don’t understand how you can make such a claim. Additional information is needed, please consider my comments / questions related to the discussion section.

Specific comments for corrections / clarifications

Abstract

The method description should be improved in the abstract. For instance, you wrote “We introduced new and accelerometer specific data augmentation methods”: could you briefly describe these new methods?

“the trained model significantly over- or underestimated time in bed.”: could you add numeric indicators showing this over or underestimation?

Materials and methods

“To ensure data quality, only participants with an intersection over the union (IoU) score of over .9 between the two annotations were included in this study.” (l 91): Please clarify this sentence. For instance, what are “the annotations”? Are they the annotators’ labeling? How was computed the score?

“Each participant was maximally included once in this subset.” (l 101): What was the criterion for choosing one of the 5 days (7 – 2 days) for each participant?

Procedure

It seems that you use two different methods to label time in bed minutes. On one hand, you use Actiwave Cardio data from the “Tromsø Study TiB dataset” to label the TiB minutes from the ActiGraph sensor data. On the other hand, you use ActiGraph data from the “Tromsø Study accelerometer dataset” to label the TiB minutes from the ActiGraph sensor data.

• Why don’t you use the same method to label the ActiGraph sensor data? It should be explained. It is important to show that no bias could result from the use of two different methods, or at least that this provide a gain in your study.

• What is the discrepancy between the two methods? This could be an important information to add in supplementary materials if you explain and highlight the importance in using these two different methods.

“Also with 50% likelihood, the sequence got reversed.” (l 126): Why did you reverse the sequence? What do you expect to simulate with this?

“As for the ActiGraph/Oura/Polar validation dataset only total sleep duration data was available, we calculated the total TiB from the predictions of our model on all fully recorded days.” (l 137): If I well understand, this means that you compared the sleep duration from ActiGraph/Oura/Polar with the time in bed from your model predictions, but these features (i.e. sleep duration and time in bed) are generally different. I think this is an important bias when using the ActiGraph/Oura/Polar validation dataset. This may explain the highlighted discrepancy in your results. Have you considered deriving total sleep time to time in bed or the reverse before proceeding the comparison?

Adding a flow figure showing the procedure and how the different datasets were used would be helpful for the reader.

Results

You provided the accuracy in your results, but what about the sensitivity and specificity?

Similarly, you provided the accuracy related to the “Tromsø Study TiB dataset” and the “Tromsø Study accelerometer dataset”, and what about the TiB labeled, TiB detected, etc.? Additional indicators (average of minima, maxima or means, etc. for instance; you could consider root mean square errors as well…) used to compare the output and the expected results would be relevant.

Discussion

Please, consider these two sentences “It remains also an open question to which extent our results generalize to the detection of sleep, which we could not test within the scope of this study as no appropriate data from hip-worn accelerometers is currently available.” (l 181) and “A crucial strength of this paper is to provide an opportunity to estimate TiB from already collected data. With this, we anticipate the chance for longitudinal analyses of sleep and TiB-related research question in settings where these questions only recently got into focus, but prior data exists.” (l 219) I am a bit confused. There seems to be a contradiction between these two statements about the availability of data on which to apply your method. Could you clarify what you mean?

“Another important contribution of this study is the introduction of accelerometer-specific data augmentation as a tool to improve model generalization.” (l 230): I am not sure that the reader well understands the point of using this technique in the procedure. As this is not the purpose of the article, could you please add a table in supplementary materials showing the quantified gain in accuracy between model basically trained and a model trained with data augmentation?

Figure S1

It would be helpful in Figure S1 to show which participant met the requirements fixed in your study, and for those retained, to describe (for instance, with another color and a thick, continuous line) which parts were considered as time in bed according to your criteria.

Figure S4

What are the characteristics of the machines: desktop and laptop (processor, RAM, 64 or 32 bits)?

Typos

Introduction (l 21): descriptopn -> description

Results (l 148, 152): hyperparamter -> hyperparameter

Discussion (l 207): over -> offer?

Discussion (l 246): a algorithm -> an algorithm

Best of luck for your paper submission.

Reviewer #2: The present study examined an important topic in the literature - the use of accelerometer to evaluate sleep. The development of an algorithm using the information of hip-worn accelerometer data would allow the exploration of a high amount of data that previously used accelerometer in the hip. Despite an important topic, the study has some important caveats that should be addressed. See my following comments that should be addressed:

ABSTRACT

The authors should better provide the rationale of the study - what is the potential advantage of measuring time in bed using hip-worn accelerometer? The objective of the study is not clear. The authors should provide more detailed results in the abstract.

METHODS - Datasets

The authors should provide a more clear and detailed description of the datasets used in the study. Why the authors choose these three different datasets?

In the Tromso Study TiB dataset, where the accelerometers were placed?

Line 79 - Why did the participants in the Tromso Study TiB dataset wear the accelerometers only during 24-hours?

Is the ActiGraph monitor cited in the Tromso Study accelerometer dataset the same than the used in the Tromso Study TiB dataset? This is not clear in the manuscript.

The idea of evaluating the accuracy of information comparing with consumer-wearable data is good. However, the very limited information of this validation dataset (n=16) is a reason of concern.

Procedure: It is not clear how the performance of the model was assessed.

A more detailed description of the Van Hees et al. method is required in the manuscript.

RESULTS

Description is poor, without presenting most of statistical estimates.

6. PLOS authors have the option to publish the peer review history of their article (what does this mean?). If published, this will include your full peer review and any attached files.

Reviewer #1: No

Reviewer #2: No

---

## [Author Response · Author response to Decision Letter 1]

23 Oct 2024

Response to Editor and Reviewers

Dear editor,

dear reviewers,

We thank you for your constructive and valuable feedback. We hope we

have addressed your concerns to your satisfaction. Please find below our

responses to each of the points raised.

Best regards

Marc Weitz

> Response to the academic editor

>

> 1\. Please ensure that your manuscript meets PLOS ONE\'s style

> requirements, including those for file naming. The PLOS ONE style

> templates can be found at

>

> https://journals.plos.org/plosone/s/file?id=wjVg/PLOSOne_formatting_sample_main_body.pdf

> and

>

> https://journals.plos.org/plosone/s/file?id=ba62/PLOSOne_formatting_sample_title_authors_affiliations.pdf.

We carefully read style templates and adjusted our manuscript

accordingly. We corrected the corresponding author marking and mail

address formatting. Please let us know if you find any other style

violations.

> 2\. We note that you have indicated that there are restrictions to data

> sharing for this study. PLOS only allows data to be available upon

> request if there are legal or ethical restrictions on sharing data

> publicly. For more information on unacceptable data access restrictions,

> please see

> http://journals.plos.org/plosone/s/data-availability#loc-unacceptable-data-access-restrictions.

>

> Before we proceed with your manuscript, please address the following

> prompts:

>

> a\) If there are ethical or legal restrictions on sharing a

> de-identified data set, please explain them in detail (e.g., data

> contain potentially identifying or sensitive patient information, data

> are owned by a third-party organization, etc.) and who has imposed them

> (e.g., a Research Ethics Committee or Institutional Review Board, etc.).

> Please also provide contact information for a data access committee,

> ethics committee, or other institutional body to which data requests may

> be sent.

There are in total three data sets used in this study: The Tromsø Study

TiB dataset, the Tromsø Study accelerometer dataset, and the

ActiGraph/Oura/Polar validation dataset. All these data sets are

secondary data that have been and can be obtained from the respective

owners. We reached out to the respective data owners regarding your

request of openness of the data. Please see the respective responses

below.

For the Tromsø Study TiB dataset and the Tromsø Study accelerometer

dataset, the legal restriction on data availability is set by the Tromsø

Study Data and Publication Committee to control for data sharing,

including publication of datasets with the potential of reverse

identification of de-identified sensitive participant information. The

data can, however, be made available from the Tromsø Study upon

application to the Tromsø Study Data and Publication Committee. Contact

information: The Tromsø Study, Department of Community Medicine, Faculty

of Health Sciences, UiT The Arctic University of Norway; e-mail:

<tromsous@uit.no>. We reached out to the Tromsø Study about the

possibility to publish the data used in this study, but because of the

legal restrictions, we expect this process to take time.

The ActiGraph/Oura/Polar validation dataset has been collected by

Henriksen et al. (2022). The authors plan to publish the data openly,

however, currently it is only available upon request to the study\'s

corresponding author. While there is an overlap between the authors of

this study and Henriksen et al. (2022), we do not own this data and have

therefore no possibility to make it openly available.

> b\) If there are no restrictions, please upload the minimal anonymized

> data set necessary to replicate your study findings to a stable, public

> repository and provide us with the relevant URLs, DOIs, or accession

> numbers. For a list of recommended repositories, please see

>

> https://journals.plos.org/plosone/s/recommended-repositories. You also

> have the option of uploading the data as Supporting Information files,

> but we would recommend depositing data directly to a data repository if

> possible.

>

> We will update your Data Availability statement on your behalf to

> reflect the information you provide.

Please see the information provided under 2a, why we cannot share the

data.

> 3\. In the online submission form, you indicated that \[The data of the

> Tromsø Study cannot be shared publicly because of the risk of potential

> reidentification, but can be obtained upon application to the Data and

> Publication Committee for the Tromsø Study:

> https://uit.no/research/tromsostudy. The ActiGraph/Oura/Polar validation

> dataset can be obtained from A.He. (andre.henriksen@uit.no) upon

> reasonable request.\].

>

> All PLOS journals now require all data underlying the findings described

> in their manuscript to be freely available to other researchers,

> either 1. In a public repository, 2. Within the manuscript itself, or 3.

> Uploaded as supplementary information.

>

> This policy applies to all data except where public deposition would

> breach compliance with the protocol approved by your research ethics

> board. If your data cannot be made publicly available for ethical or

> legal reasons (e.g., public availability would compromise patient

> privacy), please explain your reasons on resubmission and your exemption

> request will be escalated for approval.

Please see our statement under 2a, why we cannot share the data.

> Response to the reviewer #1:

>

> Reviewer #1: This paper presents an algorithm aiming at discriminating

> time in bed and time out of bed based on hip-worn accelerometry data.

> While hip-worn accelerometry data focused on physical activity, such an

> algorithm would enable to reuse this data to get other outputs related

> to sleep. This algorithm is based on an augmentation data process and

> machine learning models of type Long-Short Term Memory networks (LSTM).

> This study has potential because the procedure used well-known

> algorithms applied in similar contexts and some techniques to add noise

> or take into account potential misuse of devices (flipped sequence,

> Gaussian noise), but there are weaknesses that need to be revised before

> publication.

>

> General comments

>

> As you mentioned in the paper, wearing hip-worn accelerometers during

> 24h can be a burden for participants, especially during sleep hours

> ("wrist-placement has been considered less obtrusive and thus leads to

> fewer non-wear days"; "To decrease study participant burden, many early

> study protocols comprised the removal of the device during sleep"). This

> fact might explain the lack of data appropriate for this kind of study.

> I think you should discuss the potential of using hip-worn

> accelerometers in behavior monitoring, maybe in future studies. In

> addition, it is probably a limit to the study, not so much a scientific

> limit as a functional one. It should be mentioned.

We are now discussing the potential in the first paragraph of the

discussion and mention explicitly two of the larger studies that

currently use hip-worn accelerometers with a 24h wear protocol: The

Tromsø Study and the Nako Study. The benefits of hip-worn accelerometers

with respect to physical activity alone are discussed in the

introduction.

> In your process, you use many types of data but it is unclear from the

> description how this data is used in the procedure. For instance in the

> procedure, it is not clear which dataset was used to train the models.

> Is it the "Tromsø Study TiB dataset", "the Tromsø Study accelerometer

> dataset" or a mix of both? Please clarify the function of each dataset

> you used.

We have split the Procedure section into three subsections (Data

preparation, Model training, and Model evaluation) and clarified in each

subsection which dataset was used for which step and purpose. We further

included an overview figure in Figure 1.

> I am a bit skeptical about the use of ActiGraph/Oura/Polar dataset. Only

> 15 (16 -- 1) participants provide "relevant information". Moreover, it

> seems there is a confusion between sleep duration and time in bed

> (please consider my comments / questions related to the procedure

> section).

We fully understand the skepticism about this dataset and strongly

acknowledge its limitations. However, we believe that, despite its

limitations, this dataset addresses an important issue of this paper,

that is the generalization of the model. To avoid the impression of

overinterpreting this data, we extended the discussion of its

limitations, and also stated the objective of its use more clearly. We

also agree on the difficulty of comparing TST with TiB, but again, our

objective with this dataset was to test for \"reasonable results\" in

the sense of slightly higher but not far off TiB estimates compared to

TST. We changed the text to make this point clear.

> You mentioned that "data augmentation" improved the model

> generalization. How this technique improved the model. There is no

> comparisons with and without data augmentation. I don't understand how

> you can make such a claim. Additional information is needed, please

> consider my comments / questions related to the discussion section.

We rephrased this claim as we did not find a good way of quantifying

this as the matter is more complex than only accuracy comparisons. We

also highlighted now that our choices here were heuristically guided and

motivated by data augmentation in other disciplines such as computer

vision and natural language processing. Our intention was not to focus

this paper on this topic (even though we believe that this is a crucial

topic currently missing in the literature) and that we would have liked

to make more informed decisions about augmentation than the

heuristically motivated choices we have made. In fact, our intention was

to point out this knowledge gap in the discussion, and we hope that we

have made this clearer now. Finally, we want to point out that we are

doubtful that the question of augmentation can be solved empirically in

singular experiments. We are at least not aware of any papers in the

above-mentioned disciplines who did that. Our impression from these

disciplines is that over time some consensus/ideas of which augmentation

is reasonable on which task has emerged. So, we hope that in the

discussion, we managed to make clear now that regarding the augmentation

we are using here, we neither claim to be complete nor correct, but that

it is reasonable to apply it.

> Abstract

>

> The method description should be improved in the abstract. For instance,

> you wrote "We introduced new and accelerometer specific data

> augmentation methods": could you briefly describe these new methods?

Given the limited space in the abstract, we could only include a very

brief explanation of the augmentation methods used. For the more

interested reader, a longer description is provided in the methods

section.

> "the trained model significantly over- or underestimated time in bed.":

> could you add numeric indicators showing this over or underestimation?

Our intention with this sentence was to refer to the few cases in which

the model predicted all or nothing as time in bed (please see in Fig 1B

the points at 0 and 1440 minutes of predicted time in bed, which we

marked as \"outlier predictions\"). We added this information now in the

respective sentence and replaced significantly by fundamentally to avoid

confusion with statistical significance.

> Materials and methods

>

> "To ensure data quality, only participants with an intersection over the

> union (IoU) score of over .9 between the two annotations were included

> in this study." (l 91): Please clarify this sentence. For instance, what

> are "the annotations"? Are they the annotators' labeling? How was

> computed the score?

We now consistently use the term \"annotators\' ratings\" (as used

already a couple of times in the text) to avoid confusion. Please

indicate if this term seems misleading to you. We also added a

description of how the IoU score was computed. A visualization and the

mathematical formula are now included in S1 Fig.

> "Each participant was maximally included once in this subset." (l 101):

> What was the criterion for choosing one of the 5 days (7 -- 2 days) for

> each participant?

We are not 100% sure about the exact meaning of this question but

anticipate that our description of the sampling procedure is confusing.

For that reason, we added more information about the accelerometer data

in the Tromsø Study (\~6000 participants with 7 days recording each) and

that we sampled 500 days from these \~42000 days to have a feasible

sample of approximately the same size as the TiB dataset providing

additional heterogeneity (e.g., people with potentially different

sleeping behavior).

> Procedure

>

> It seems that you use two different methods to label time in bed

> minutes. On one hand, you use Actiwave Cardio data from the "Tromsø

> Study TiB dataset" to label the TiB minutes from the ActiGraph sensor

> data. On the other hand, you use ActiGraph data from the "Tromsø Study

> accelerometer dataset" to label the TiB minutes from the ActiGraph

> sensor data.

>

> • Why don't you use the same method to label the ActiGraph sensor data?

> It should be explained. It is important to show that no bias could

> result from the use of two different methods, or at least that this

> provide a gain in your study.

>

> • What is the discrepancy between the two methods? This could be an

> important information to add in supplementary materials if you explain

> and highlight the importance in using these two different methods.

The method for the TiB and the accelerometer dataset is largely the same

except that for the TiB dataset, we had an additional sensor (the

ActiWave Cardio) available which we utilized for this dataset in

addition. The ActiWave Cardio provided additional and reliable

inclination data as it was glued to the participants' chest. This

additional inclination data is also visualized now in supplementary

material S1. For the accelerometer dataset only the ActiGraph was

available. We now clarified this in the respective dataset descriptions.

We also added a brief statement about our motivation for the Tromsø

Study accelerometer dataset and highlighted the similarities (same

overall labeling procedure) and differences (Availability of the

ActiWave Cardio data only in the TiB dataset) to avoid confusion.

> "Also with 50% likelihood, the sequence got reversed." (l 126): Why did

> you reverse the sequence? What do you expect to simulate with this?

We added our intention that this can simulate inverse turning behavior

(e.g. adding a left-to-right turn in the night from a right-to-left

turn)

> "As for the ActiGraph/Oura/Polar validation dataset only total sleep

> duration data was available, we calculated the total TiB from the

> predictions of our model on all fully recorded days." (l 137): If I well

> understand, this means that you compared the sleep duration from

> ActiGraph/Oura/Polar with the time in bed from your model predictions,

> but these features (i.e. sleep duration and time in bed) are generally

> different. I think this is an important bias when using the

> ActiGraph/Oura/Polar validation dataset. This may explain the

> highlighted discrepancy in your results. Have you considered deriving

> total sleep time to time in bed or the reverse before proceeding the

> comparison?

We acknowledge that these are fundamentally different measures, and it

is very hard/impossible to compare them. However, please note that as we

have pointed out now more clearly in the text, the motivation to include

this dataset was to primarily obtain a better idea on whether the

model\'s predictions are reasonable on demographically different data

than the data it was trained on. This information could not be gained by

the TiB dataset as it, even though its different data than in the

training, shares comparable demographic characteristics. We also want to

note, that this data is, without any doubt, not the most ideal data and

has many

---

## [Decision Letter · Decision Letter 1]

9 Mar 2025

Automatic time in bed detection from hip-worn accelerometers for large epidemiological studies: The Tromsø Study

PONE-D-24-15603R1

Dear Dr. Weitz,

We’re pleased to inform you that your manuscript has been judged scientifically suitable for publication and will be formally accepted for publication once it meets all outstanding technical requirements.

Kind regards,

Julio Alejandro Henriques Castro da Costa

Academic Editor

PLOS ONE

Additional Editor Comments (optional):

Reviewers' comments:

Reviewer's Responses to Questions

**Comments to the Author**

1. If the authors have adequately addressed your comments raised in a previous round of review and you feel that this manuscript is now acceptable for publication, you may indicate that here to bypass the “Comments to the Author” section, enter your conflict of interest statement in the “Confidential to Editor” section, and submit your "Accept" recommendation.

Reviewer #1: All comments have been addressed

2. Is the manuscript technically sound, and do the data support the conclusions?

Reviewer #1: Yes

3. Has the statistical analysis been performed appropriately and rigorously? 

Reviewer #1: Yes

4. Have the authors made all data underlying the findings in their manuscript fully available?

Reviewer #1: No

5. Is the manuscript presented in an intelligible fashion and written in standard English?

Reviewer #1: Yes

6. Review Comments to the Author

Reviewer #1: Is the manuscript technically sound, and do the data support the conclusions?

Yes.

Has the statistical analysis been performed appropriately and rigorously?

Yes.

Have the authors made all data underlying the findings in their manuscript fully available?

No. However, the authors explained why in the dedicated section.

Is the manuscript presented in an intelligible fashion and written in standard English?

Yes. I am not an English speaker but I think it is clear.

Review Comments to the Author

General Comments

The current manuscript is much clearer than the first one. The authors adequately addressed the comments I previously raised.

I only have a last suggestion regarding lines 181, 182, 186: Please write specificity and sensitivity in percentages (like accuracy for homogeneity).

All the best in the process.

7. PLOS authors have the option to publish the peer review history of their article (what does this mean?). If published, this will include your full peer review and any attached files.

Reviewer #1: No

---

## [Editor Report · Acceptance letter]

PONE-D-24-15603R1

PLOS ONE

Dear Dr. Weitz,

I'm pleased to inform you that your manuscript has been deemed suitable for publication in PLOS ONE. Congratulations! Your manuscript is now being handed over to our production team.

Kind regards,

on behalf of

Dr. Julio Alejandro Henriques Castro da Costa

Academic Editor

PLOS ONE